# Design and Implementation of a Co-Simulation Framework for Testing of Automated Driving Systems

**Demin Nalic [1,*] , Aleksa Pandurevic [1], Arno Eichberger [1] and Branko Rogic [2]**

[1]  Institute of Automotive Engineering, TU Graz, 8010 Graz, Austria; pandurevic@tugraz.at (A.P.); arno.eichberger@tugraz.at (A.E.)

[2]  MAGNA Steyr Fahrzeugtechnik AG Co. & KG, 8045 Graz, Austria; branko.rogic@magna.com

*  Correspondence: demin.nalic@tugraz.at

**Abstract:** The increasingly used approach of combining different simulation softwares in testing of automated driving systems (ADSs) increases the need for potential and convenient software designs. Recently developed co-simulation platforms (CSPs) provide the possibility to cover the high demand for testing kilometers for ADSs by combining vehicle simulation software (VSS) with traffic flow simulation software (TFSS) environments. The emphasis on the demand for testing kilometers is not enough to choose a suitable CSP. The complexity levels of the vehicle, object, sensors, and environment models used are essential for valid and representative simulation results. Choosing a suitable CSP raises the question of how the test procedures should be defined and constructed and what the relevant test scenarios are. Parameters of the ADS, environments, objects, and sensors in the VSS, as well as traffic parameters in the TFSS, can be used to define and generate test scenarios. In order to generate a large number of scenarios in a systematic and automated way, suitable and appropriate software designs are required. In this paper, we present a software design for a CSP based on the Model–View–Controller (MVC) design pattern as well as an implementation of a complex CSP for virtual testing of ADSs. Based on this design, an implementation of a CSP is presented using the VSS from IPG Automotive (CarMaker) and the TFSS from the PTV Group (Vissim). The results showed that the presented CSP design and the implementation of the co-simulation can be used to generate relevant scenarios for testing of ADSs.

**Keywords:** ADAS simulation; scenario generation; automated driving; testing; innovation in mobility; self-driving cars; transportation

## 1. Introduction

Testing and validation procedures are essential for the safety approval and acceptance of automated driving systems (ADSs). Recent studies have shown that the main concerns regarding the acceptance of ADSs in Austria are reliability and safety (see [1]). For ADSs with Level 3+ automation levels (defined as in [2]), the testing and validation procedures are very complex. The high demand for testing kilometers as well as real-world testing (RWT) is time- and resource-consuming (see [3–5]). In addition to the costs and time demands of RWT, the future technologies presented in [6] and impact analyses of ADSs in traffic [7,8] are very difficult to implement and test in the real world. To illustrate the complexity of RWT for a special scenario including several participants, we present an example of an intersection in Figure 1. In it, we have three different cars and two pedestrians placed in the intersection.

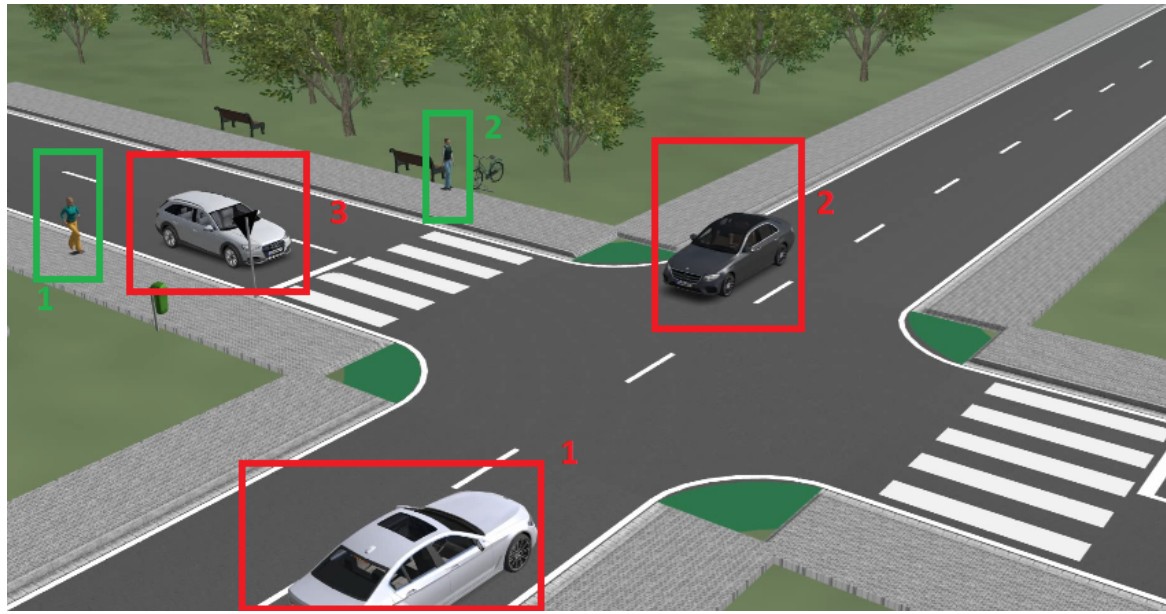

(**a**) Intersection scenario with three vehicles in red rectangles and two pedestrians in green rectangles.

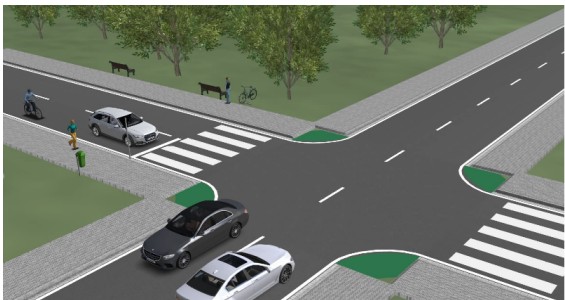

(**b**) Time frame 1–vehicle 2 is crossing the street while the other vehicles are waiting. Pedestrian 1 is moving to the zebra crossing and pedestrian 2 is standing still.

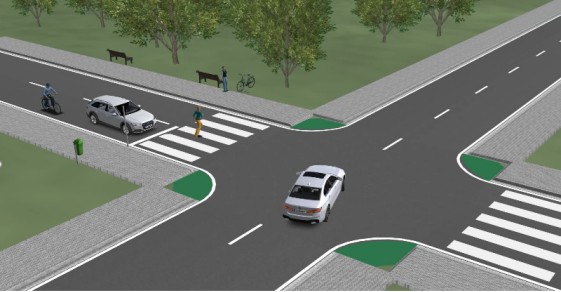

(**c**) Time frame 2–vehicle 1 starts moving to the left toward the zebra crossing while vehicle 3 is standing still. Pedestrian 1 is in the middle of the the zebra crossing and pedestrian 2 is standing still.

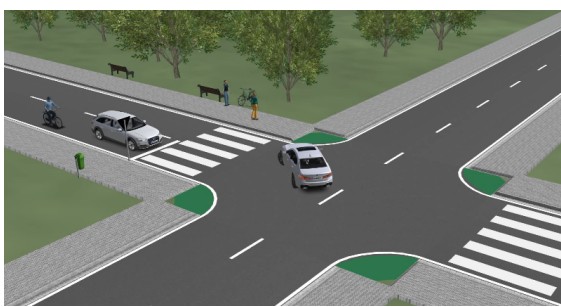

(**d**) Time frame 3–vehicle 1 is moving very close to the zebra crossing. Pedestrian 1 has crossed the zebra crossing and pedestrian 2 is standing still.

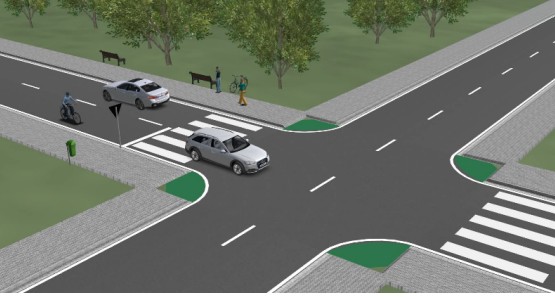

(**e**) Time frame 4–vehicle 1 continues driving on the new lane and vehicle 3 starts driving in the initial lane.

**Figure 1.** Intersection scenario with five objects, three vehicles and two pedestrians. The cyclist which occurs in time frame 1 is not considered for the actual scenario calculation.

If we define parameters with defined ranges for all intersection participants, it is possible to generate different scenarios on this intersection by combining these parameters. In this case, an exemplary number of parameters and parameter ranges with five elements are defined. The parameters that are defined are the vehicle initial velocities $v_{veg}^i \in [10 \text{ km/h}, 15 \text{ km/h}, 20 \text{ km/h},$

30 km/h, 40 km/h], the vehicle routes $r^i_{veh} \in [1,2,3,4,5]$, the lateral displacement from the vehicle route $r^i_{lat} \in [0.25$ m, 0.5 m, 1.0 m, 1.5 m, 2.0 m], the pedestrian routes $r^j_{ped} \in [1,2,3,4,5]$, and the initial pedestrian velocities $v^j_{ped} \in [0$ km/h, 2.5 km/h, 5 km/h, 7.5 km/h, 10 km/h], with the index $i$ representing the number of vehicles and index $j$ representing the number of pedestrians in the intersection in Figure 1. By calculating all possible combinations of these parameters, we could create 3125 scenarios. If one scenario takes 10 min in the real world, we would need 31,250 min of RWT, which is around 22 days of RWT. Considering higher parameter ranges and more participants, it would jump to an infinite number of scenarios for these specific intersections. Comparing this with a continuous simulation where one such scenario in CarMaker could be calculated in 5 s, all scenarios could be calculated and tested in 52 min assuming these parameter ranges. It should be mentioned here that even if it is possible to reduce the calculation and generation time of scenarios, not all combinations are relevant in full factorial testing for the system being tested. To reduce the number of scenarios to only relevant scenarios, the combinatorial testing methods presented in the research works of [9–11] can be used. By using combinatorial testing, it is possible to define fewer parameters and parameter interactions where not all parameter combinations are taken into account. Using suitable virtual platforms and simulation tools for generation of test scenarios in testing of ADS is a way to reduce and overcome RWT. To ensure representative and valid data from virtual platforms, those platforms should provide the possibility to implement and develop realistic and complex vehicle, traffic, object, sensor, and environment models, as well as other real-world elements that are relevant for the vehicle being tested. These virtual platforms can be found in the research works presented in [12–14]. The platforms have a common quality, they combine vehicle simulation software (VSS) with a traffic flow simulation software (TFSS) in a co-simulation. Such co-simulation platforms (CSPs) provide the opportunity to test the ADS in an approximately realistic environment, thus reducing the efforts in RWT. In this paper, we present an approach where, with the combination of a VSS with a TFSS, a vehicle under test is tested in a stochastic traffic environment in order to create relevant scenarios. The stochastic traffic environment provides the possibility of detection of scenarios that cannot be easily found through exhaustive combinatorial testing. The stochastic traffic environment corresponds to approximately real test-driving conditions, where a vehicle under test is tested as it would be in RWT procedures. To create such an environment, the approach uses two components, one for modeling of the ego vehicle with its automated driving functions, and another for modeling the traffic flow. For the vehicle simulation, the widely used software in the automotive engineering community is CarMaker, and for the traffic flow simulation, Vissim, presented in [15], is used. The CarMaker VSS provides a virtual multi-body simulation that is a computer-modeled representation of a vehicle (see [16]). Additionally, it provides the possibility to implement ADSs, sensor models, adjustments, and models of objects in the virtual environment of CarMaker. The traffic flow model in Vissim is calibrated by using data measured in an official test road in Austria [17], and is presented in [12]. The concept and the modeling part of CSP, the vehicle under test, and the traffic environment are also presented in [12]. The main goal of this paper is to show and present a design of a CSP with an implementation using a co-simulation combining CarMaker and Vissim for the generation of scenarios for testing ADSs (e.g., [18,19]). The results and efficiency of the presented co-simulation framework are summarized in Section 5.

## 2. Co-Simulation Framework

The high-level over of the co-simulation framework (CSF) can be seen as a group of three components interacting with each other, as shown in Figure 2.

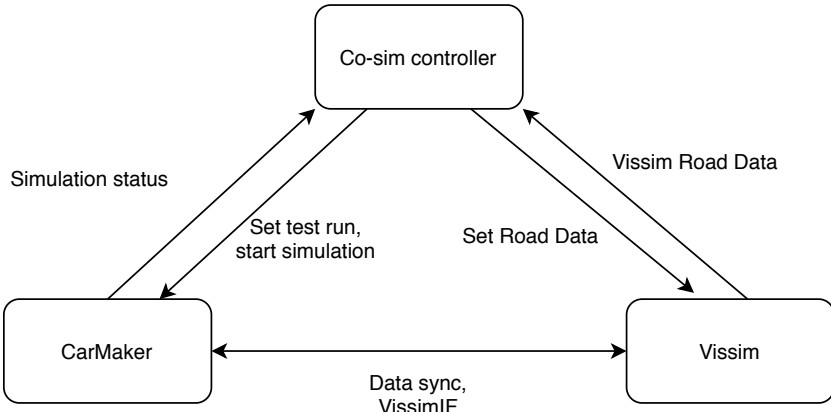

**Figure 2.** A high-level overview of the software architecture presenting the three components of the co-simulation framework.

In a broad sense, with these existing components, the remaining tasks are the development of the co-simulation controller (CSC) and the communication between the CSC, CarMaker, and Vissim. Communication between Vissim and CarMaker is handled through the Vissim Interface for CarMaker (see [20]). Vissim provides a useful COM interface through which the communication between the CSC and Vissim is handled (see [21,22]). Communication between the CSC and CarMaker is done in two ways: by sending *Tcl* commands through the *cmguicmd* interface or by setting the Simulink model parameters pragmatically. Considering the whole framework, most of the development is done regarding the CSC. Using object-oriented concepts, data and the behaviors related to those data are coupled together. Another goal of introducing object-oriented concepts was a separation of logic from the user-interface-related code.

## 3. Software Design

Software design was the essential part of this research. The CSC is broken into multiple components, where each is described with at least one class without violating any of the functional requirements.

### 3.1. Defining Functional Requirements

The functional requirements for the software design are shown in Table 1.

**Table 1.** Functional requirements of the co-simulation framework.

| | |
|---|---|
| Requirement 1 | Simulation |
| Requirement 2 | Data Storage |
| Requirement 3 | Road File |
| Requirement 4 | Simulation Control |
| Requirement 5 | Input Validation |

The main functional requirement in Table 1 is the first one. It is divided into additional simulation parts, the simulation of one and multiple scenarios over many kilometers.

3.1.1. Simulation of One Specific Scenario over Many Kilometers

The co-simulation framework provides the possibility to choose the test road, the Vissim version, the testing kilometers, and the specific scenario test run for scenario-generation purposes. In CarMaker, a scenario test run is a specified simulation run that comprises the vehicle under test with all functionalities, a defined test road, environment objects, and a test definition. All given parameters are checked by the system for validity before starting the simulation. A scenario test run is a test run defined with a configured vehicle model and traffic simulation.

### 3.1.2. Simulation of Multiple Scenarios over Many Kilometers

The co-simulation framework provides the possibility to choose the root folder that contains scenarios, and a selection of those which should be simulated. Each scenario test run is simulated over a given number of kilometers. All given parameters are checked by the system for validity before starting the simulation.

### 3.2. Design

For the design of the co-simulation framework, a graphical user interface (GUI) is implemented using a model-view-control (MVC) pattern [23]. By applying MVC pattern, elements of the view component are decoupled from the framework's logic, and the rest of the code is broken into small yet meaningful components, which interact with each other. For the implementation of the MVC design pattern, the guidelines from [24] were used. The interactions between components can be seen in Figure 3.

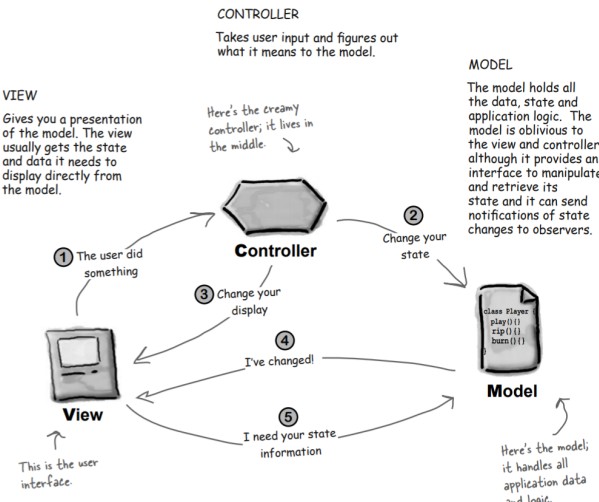

**Figure 3.** Representation of the interactions between components of the MVC (image taken from [24]).

In the co-simulation framework, these three components can be distinguished as *View*, *Controller*, and *Model*. The *View* component is of the *singleton View* class, which describes the appearance of the user interface. All user interface elements are defined there. The *Controller* component is of the *singleton Controller* class, which acts as an interface between the *View* and *Model* components. It updates the state of the *Model* component, handles return values of the *Model* functions, and, based on them, generates an adequate graphic response using *View* elements. Compared to the previous two components, the *Model* component is not a single class. There is a *singleton Model* class, but it is just the main part of the model component. The *Model* component is composed of additional classes that are responsible for the framework logic and for holding all necessary data for the simulation process. The control logic of framework is located in the *StartSimulation* function. It differentiates between two different modes of operation, the manual mode and the cluster mode. The model class is also responsible for the inter-process communication with Vissim and CarMaker. When the *StartSimulation* function is called, the *Model* component configures both Vissim and CarMaker and executes simulations.

### 4. Application

Two separate modes of simulations can be distinguished from functional requirements, the manual mode and automatic or cluster mode (see [12]). In the manual mode, the user is allowed to generate a very specific scenario where they can measure different parameters in the tool, such as distributions,

compositions of vehicles, and traffic. The automatic or cluster mode is meant for exhaustive fully automated testing of a model over multiple kilometers.

*4.1. Cluster Mode*

The automatic or cluster mode of the co-simulation is a simulation of predefined scenarios created using real traffic measurements of a specific road (see [12]). Using the data provided, the user wants to observe the behavior of the modeled vehicle over varying driving distances. In this mode, the user can choose the path to the root folder, where clusters containing predefined scenarios are located. All clusters found in the selected root folder are displayed to the user in the GUI and are made available for selection, as presented in Figure 4. Execution of cluster mode can be separated into two phases, the initialization phase and simulation phase. The initialization phase comprises:

1. Loading the CarMaker testrun file
   The file with the specified direction and the Vissim road file are loaded into CarMaker.
2. Loading the Vissim road file
   The road from the selected cluster is loaded into Vissim, and it is modified by changing the random seed value. The random seed value changes the traffic procedure without changing the parameterization. This value represents the stochastic nature of the road traffic.
3. Setting the number of kilometers to be covered
   The number of kilometers specified by the user is set in the Simulink model using the *set_param* function.
4. Starting simulation
   The simulation is started by setting the *SimulationCommand* parameter of the Simulink model to *start*. This can also be done using the *set_param* function.

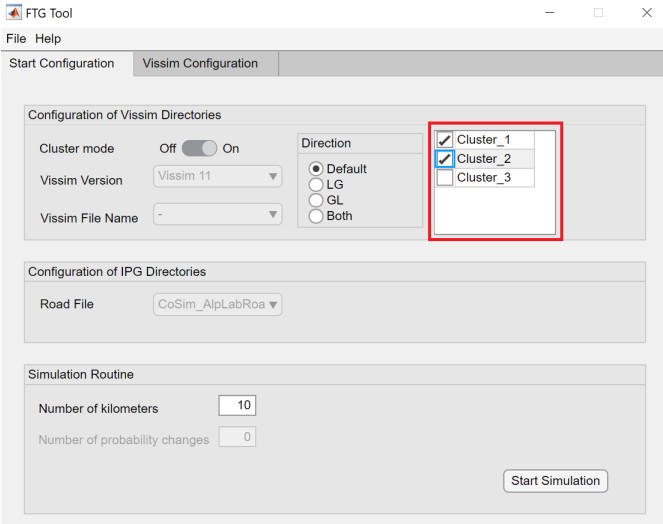

**Figure 4.** Selecting clusters loaded from the cluster's root path in the graphical user interface (GUI).

At this point, the simulation phase begins, and the status of the running simulation has to be checked. For this purpose, timers are used, as described in Section 4.3. If the simulation is stopped, the covered kilometers are checked. If the simulation status reaches a given number of kilometers, the CSC can proceed in three following ways:

1. Restarting the simulation on the same road but in the opposite direction.
2. Switching to the following cluster.
3. Finishing if no more clusters are left.

During the simulation phase, the progress bar is presented to the user with three purposes: The first is so that the user is aware of the running simulation's status. The second purpose

is the possibility of cancelling the simulation via the stop button, which is part of the progress bar. The displayed progress bar deactivates the whole user interface so that no changes of the simulation settings can be made until the stop button is pressed, and that is its third purpose. The obvious advantage of this mode is the automation of a process that, if done manually, would consume more time. The data generated by Vissim and CarMaker are stored in such way that they can be accessed by date, mode, or creation time stamp. Post-processing is done at the end, where data for all clusters are acquired. Together with all generated data, the configuration file in which the configuration for the latest simulation run can be found, including the cluster root folder path, selected clusters, number of kilometers per cluster, and random seeds used for every simulation step is stored.

*4.2. Manual Mode*

The main difference between the manual and cluster modes is that in the manual mode, the user does not use predefined clusters for the simulation, but, rather, individual Vissim road files that can be easily modified through the GUI. All the data related to the chosen Vissim road file are loaded into the internal data structure using the COM interface and are displayed in a separate tab in the GUI, as presented in Figure 5. Vissim-related data are held by the *VissimData* class, which, when initialized, reads the data from Vissim and has the responsibility of filling the data structure. The process of loading the data is done in the *VissimData* constructor, which is called upon the selection of the Vissim file. All operations on the data are handled by the *VissimData* class. Using this class provides convenient extraction of the Vissim data. It also enables having multiple Vissim files loaded into the memory. Switching between them is a lot faster than loading all the data all over again. When all the data from Vissim are retrieved, the data are placed into the structures. Lists of compositions and volumes are created, where both lists are contained within *VissimData* class. Each volume has a reference to the corresponding composition so it can keep track of the current composition that is assigned to that particular volume. From this point, the CSF is ready for the simulation run; again, code execution after pressing the Start Simulation button is separated into two phases, the initialization and simulation phases. At the beginning of the initialization phase, all changes of compositions and volumes are sent back to Vissim, which is also the main difference between the cluster and manual modes in this phase. Changes that user is able to perform are as follows:

1. Changing the vehicle input volume value;
2. Changing the vehicle input composition;
3. Modifying the composition by modifying vehicles of that composition;
4. Changing the speed distribution of the selected vehicle;
5. Changing the type of the selected vehicle;
6. Changing the vehicle percentage within the selected composition.

The rest of the initialization phase of the manual mode is similar to the one in the cluster mode. In the simulation phase, the main difference from the cluster mode happens when all given kilometers are covered. The simulation does not continue with a different road file, since there are no clusters to follow. The only case when the simulation continues after a selected number of kilometers is when the direction option is set to both. All performed changes are immediately reflected in the internal data structure, which is loaded into the Vissim road file upon simulation start via the interface. All the data generated by Vissim and CarMaker are stored in the same way as described in the cluster mode.

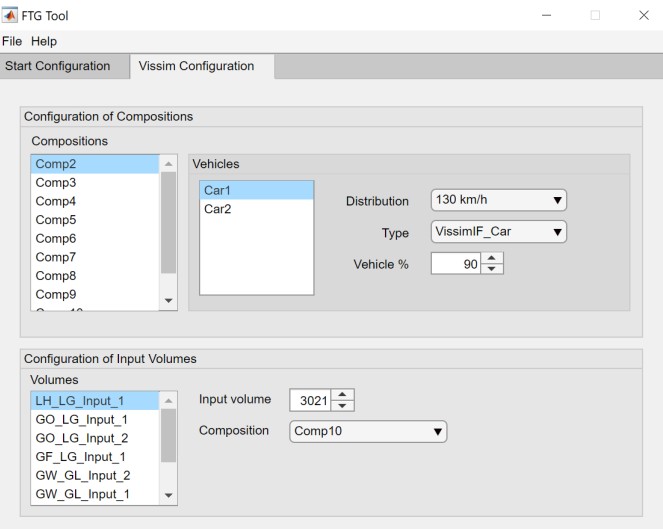

**Figure 5.** Vissim configuration tab.

### 4.3. Simulation Status

Since MATLAB lacks conventional multi-threading capabilities, timers are used as a workaround to make the GUI responsive while the simulation is running. After the simulation is initialized, the timer is started. It schedules the execution of a MATLAB code in this case, a code that checks the current simulation status. It checks whether the simulation is stopped or still running. In the case that the simulation has stopped, the number of covered kilometers is checked, and it is decided if more kilometers have to be simulated in this configuration or not. The rest of the simulation flow is specific to the mode. Since the scheduling of the code that checks simulation status is not time-critical, the *fixedSpacing* mode is chosen for the timer. This mode avoids any possibility of interrupting the execution of a function during its execution. No new execution is scheduled until the function is executed, and that plays out as an advantage, since the goal is to avoid frequent checks of the status, which are time-consuming and interrupt the simulation. It is also not desirable to fill a queue with scheduled function calls, since the simulation can be stopped at any time. As a result of using timers, the CSF has a responsive GUI, allowing the user to stop the simulation, and even more importantly, it enables the checking of the simulation status efficiently, since there is no reliable option for triggering events upon the end of the simulation, and repeatedly checking the status with delays in between keeps the system busy when not needed.

### 4.4. Covered Kilometers

Checking the covered kilometers is a common mechanism for both modes. It enables the simulation to run for any given length, even though the road file is limited to some extent. As can be seen in Section 4.3, the covered kilometers are checked periodically in the function executed by the timer. From the number of currently covered kilometers, a new number of remaining kilometers is calculated. The workflow of this procedure can be seen in Figure 6. There is no way to capture a route length that will be driven; therefore, a number of kilometers is given as the initial distance to be covered. If the remaining distance is shorter than the current route length, the vehicle will drive the remaining number of kilometers and then stop. In the case that the remaining distance is longer than the route, the simulation will stop when the end of the route is reached, and the covered distance will be saved in the workspace. That addition in Simulink is necessary in order to obtain a precise number of kilometers covered by the model vehicle.

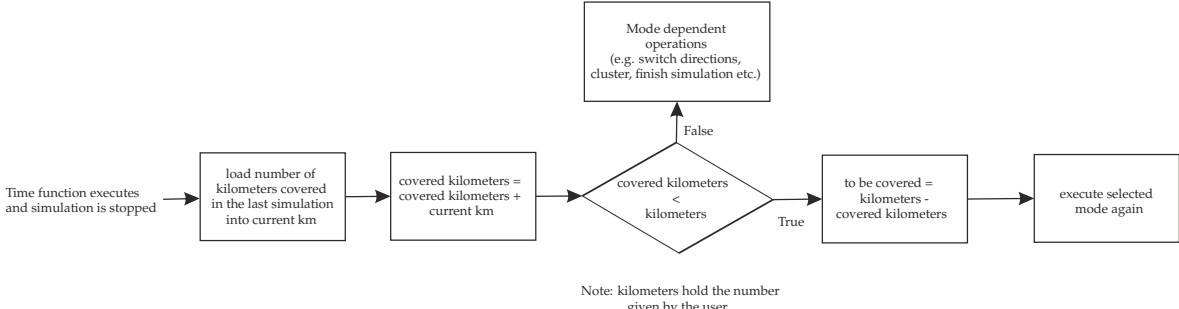

**Figure 6.** Control flow diagram showing the calculation of the remaining kilometers.

### 4.5. Communication with CarMaker

The communication with CarMaker is done in two ways, through the *cmguicmd* interface provided by CarMaker in order to send *Tcl* commands [25] and by setting the Simulink model properties using the *set_param* function (see Table 2).

**Table 2.** Different approaches of interacting with CarMaker.

| Method | Context |
|---|---|
| cmguicmd | Setting distance to be covered |
| set_param | Starting and stopping simulation |

In addition to these two options, certain parameters are set using additional scripts, such as for the driving direction in the simulation, the path in which generated data will be stored, and the Vissim road path that should be applied. Since there is no clear interface for modifying any of these parameters, scripts that modify the necessary configuration files are introduced. They are simple C++ scripts that match a given attribute name and assign a new value to it.

### 4.6. Communication with Vissim

Interaction with PTV's Vissim is completely executed through the COM interface [26]. It is used for the tasks described in Table 3.

**Table 3.** List of tasks for the interaction with Vissim.

| | |
|---|---|
| Task 1 | Setting the newly generated random seed value. |
| Task 2 | Loading compositions, vehicles, vehicle distributions, vehicle types, vehicle percentages, and vehicle input volumes. |
| Task 3 | Setting modified compositions, volumes, and vehicles. |

Before any of these operations are done, the creation of a new *actxserver* is needed, which, as an argument, requires a programmatic identifier (ProgID). It creates a COM server through which it is possible to communicate with Vissim. Depending on the Vissim version, different programmatic identifiers are passed. After starting *actxserver* and starting Vissim, the desired road network has to be loaded in order to make any changes.

### 4.7. Direction Selection

The direction in which the modeled vehicle will move is specified as an attribute in the testrun file. Since this is specific to the selected road, it is meant to be used only with predefined roads and clusters. Differences between the same test runs in different directions are not huge; therefore, the scripts mentioned in Section 4.5 are used to set certain attributes to the specified value. In addition to the two directions in which simulations can take place, there is also the possibility of choosing both directions.

In this case, the simulation runs in both directions, and it will cover the same specified number of kilometers in both of them.

*4.8. Data Storage*

The result of a successful simulation is the generation of data that are used for post-processing and data extraction. For every simulation step, one *\*.erg* file is created by CarMaker, which contains data from various ego vehicle sensors. In addition to the *erg* files, one configuration file is created, which contains all parameters used for simulation. Upon the simulation start of either of the two modes, a folder named according to the current date is created, e.g., "12-Feb-2020". Inside of it, another folder named according to the current mode is created, either "Automatic" or "Manual". All the data created by simulations on that day will be stored in the same root folder named with that date, unless the user changes the data output path. Assuming that user does not change the data output path, all automatic mode simulations would be stored in *../12-Feb-2020/automatic/* and all manual mode simulations would be stored in *../12-Feb-2020/manual/*. To avoid mixing all the data, every simulation run gets its own folder named according to the time of the simulation start, e.g., "14-30-24", in which the *erg* files and the configuration file are placed. All *erg* files and corresponding info files are simply named according to numbers that define the order of their creation.

## 5. Simulation Results

To demonstrate the effectiveness of the co-simulation framework, we used the Level3+ Highway Chauffeur presented in [12] as the vehicle under test. For that purpose, one simulation run with 5000 km was carried out. Focusing on safety-critical scenarios, collisions and near-collisions were taken for the evaluation of the framework's efficiency. Within these 5000 km of simulation, 13 collisions and 658 near-collisions were detected. Each of these detected scenarios belonged to the same defined scenario type, but had unique trajectories, vehicle states, and surrounding areas with traffic participants. As an example, two scenarios of the near-collision scenario type are presented in Figure 7. In the first panel of Figure 7a, we can see a near-collision that is caused by a vehicle in front while changing lanes in front of the vehicle under test. Figure 7b shows a near-collision caused by a truck in front of the vehicle under test while reducing its speed.

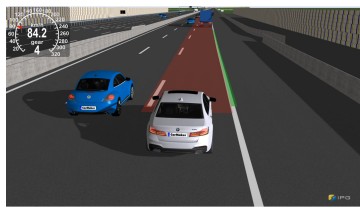 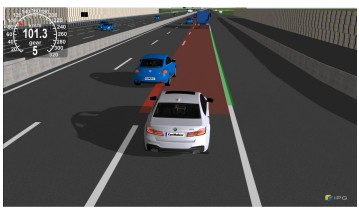 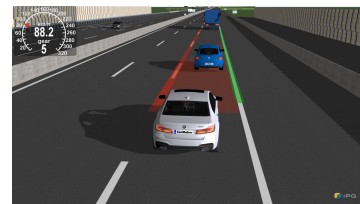

(**a**) Near-collision caused by a lane change.

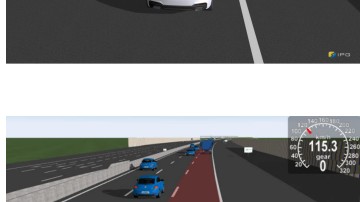 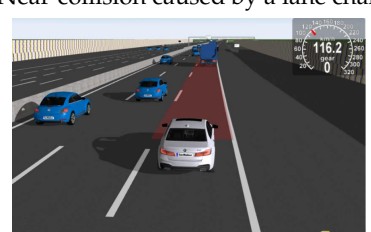 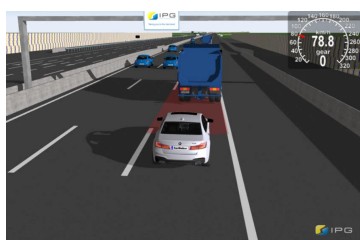

(**b**) Near-collision caused by speed reduction of the vehicle in front.

**Figure 7.** Two detected near-collisions with different causes. The vehicle under test is the white vehicle, and the blue vehicles are traffic participants generated by Vissim.

## 6. Conclusions

In this paper, we presented the design and implementation of a co-simulation framework for testing ADSs using MATLAB/Simulink, CarMaker, and Vissim. As described in Section 3, all functional requirements were defined and implemented. The results show that the implemented co-simulation framework is capable of acquiring data for scenario generation in ADSs. Since the framework highly relies on CarMaker, Vissim, and the interfaces that they provide, the implementation cannot eventually be passed to another CSP. The chosen MVC pattern for the design of the co-simulation framework was proven as an efficient and convenient way to structure a CSP and can generally be used for all CSPs that combine a VSS and TFSS.

**Author Contributions:** Conceptualization, D.N.; methodology, D.N. and A.P.; software, D.N. and A.P.; validation, A.E., B.R., and D.N.; formal analysis, D.N. and B.R.; investigation, D.N.; resources, D.N. and B.R.; data curation, D.N.; writing-original draft preparation, D.N. and A.P.; writing—review and editing, A.E. and B.R.; visualization, D.N.; supervision, A.E. and B.R.; All authors have read and agreed to the published version of the manuscript.

**Funding:** This work is funded by the Austrian Federal Ministry of Transport, Innovation, and Technology as part of the FFG Program "EFREtop".

**Acknowledgments:** Open Access Funding by the Graz University of Technology

**Conflicts of Interest:** The authors declare no conflict of interest.

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
