# Peer review of "Design and Implementation of a Co-Simulation Framework for Testing of Automated Driving Systems"

_sustainability, doi:10.3390/su122410476_

Round 1

Reviewer 1 Report

The simulation methodology is clearly explained, but no results are shown, that can prove the feasibility of the methodology.

If available, some results must be added

Author Response

Dear Mr. Reviewer,

thanky you for your Review and helpful inputs. As you suggested, I have added results of the presented method as an additional Chapter.

Best regards,

Demin Nalic

Reviewer 2 Report

Please read the attached pdf. 

Author Response

Dear Mr. Review,

Thank you for your helpful comments and inputs. In the following points, I will answer point by point the comments of yours and my adjustment to these.

  1. Abstract, Introduction and literature Review:
  2. Need to add methodology in a more elaborative way and discuss implications in the abstract.

Answer: I have extended the abstract with additional motivation about the need CSP and more explanation of the methodology by adding several sentences and the literature you suggested.

  1. There is need to expend the introduction and highlight the research gaps and how this paper contributed in overcoming those gaps.

Answer: I have adjusted the introduction in that way, that one example is given which emphasizes the need for CSP and also shows disadvantages of common ways how scenarios can be generated. Therefore I have added 3 more literature references.  I have also described in more detail what the advantage of my CSO approach is.

  1. In introduction section, I would suggest adding a clear problem with current systems

Answer:  I have added an example, which shows the problems of RTW’s and existing scenario generation methodologies, which are based on combinatorial testing.

  1. It would be nice to include some scenarios or examples to make reader understand the real word challenges

Answer:  As already mentioned in the answer b and c. An example is added and the RWT challenges and problems are emphasized and in addition there is one chapter where a detected scenario is presented using the implemented CSP.

  1. It is highly recommended to limitation and implications of findings. I would suggest
    these to be added in the conclusion and also briefly in the abstract. It would be also nice to see some potential applications scenario or examples of the real world use.

Answer:  I have added an additional chapter, which shows the results and implications using the presented design for the co-simulation framework. The Detection and generation of safety critical scenarios as collision and near-collision are evaluated using the framework. One example is given in Figure 7 where two near-collision scenarios, which were detected, are shown.

  1. English spelling and grammer.

Answer: We have checked the English grammer and spelling problems of the paper. They should be correct now.

Round 2

Reviewer 1 Report

OK, new manuscript version has been improved